# Post-Thaw Storage Temperature Influenced Boar Sperm Quality and Lifespan through Apoptosis and Lipid Peroxidation

**DOI:** 10.3390/ani14010087

**Published:** 2023-12-26

**Authors:** Junwei Li, Juncheng Li, Shuaibiao Wang, Huiming Ju, Shufang Chen, Athina Basioura, Graça Ferreira-Dias, Zongping Liu, Jiaqiao Zhu

**Affiliations:** 1College of Veterinary Medicine, Yangzhou University, Yangzhou 225009, China; lijunwei@yzu.edu.cn (J.L.); mz120211584@stu.yzu.edu.cn (J.L.); hmju@yzu.edu.cn (H.J.); 2Jiangsu Co-Innovation Center for Prevention and Control of Important Animal Infectious Diseases and Zoonoses, Yangzhou University, Yangzhou 225009, China; 3DanAg Agritech Consulting (Zhengzhou) Co., Ltd., Zhengzhou 450046, China; billwang@danagintl.com; 4Royal Veterinary College, London NW1 0TU, UK; 5Ningbo Academy of Agricultural Science, Ningbo 315040, China; jhynku@163.com; 6Department of Agriculture, School of Agricultural Sciences, University of Western Macedonia, 53100 Florina, Greece; abasioura@uowm.gr; 7CIISA—Centre for Interdisciplinary Research in Animal Health, Faculty of Veterinary Medicine, University of Lisbon, 1300-477 Lisbon, Portugal; gmlfdias@fmv.ulisboa.pt; 8Associate Laboratory for Animal and Veterinary Sciences (AL4AnimalS), 1300-477 Lisbon, Portugal

**Keywords:** boar semen cryopreservation, post-thaw storage, sperm survival, apoptosis, oxidative stress

## Abstract

**Simple Summary:**

Frozen-thawed boar semen has not been widely used due to the deleterious effects of freezing on sperm quality and fertility. Extensive attempts have been made to improve post-thaw sperm quality because the short lifespan of frozen and thawed boar sperm impedes its application in field. The present study aimed to find the best post-thaw storage method (storage time and temperature) to extend sperm survival time without affecting the quality. The results showed that post-thaw storage time and temperature influenced boar sperm survival. Storage of thawed boar semen at 17 °C preserved sperm quality better and maintained sperm quality for up to 6 h. This demonstrated an improvement in the preservation of frozen-thawed boar semen before their use for artificial insemination because it provided enough time to prepare the doses and the transport them from the laboratory to the farm. These findings will help to promote the application of frozen-thawed boar semen doses for artificial insemination in the swine industry.

**Abstract:**

Cryopreservation deteriorates boar sperm quality and lifespan, which restricts the use of artificial insemination with frozen-thawed boar semen in field conditions. The objective of this study was to test the effects of post-thaw storage time and temperature on boar sperm survival. Semen ejaculates from five Landrace boars (one ejaculate per boar) were collected and frozen following a 0.5 mL-straw protocol. Straws from the five boars were thawed and diluted 1:1 (v:v) in BTS. The frozen-thawed semen samples were aliquoted into three parts and respectively stored at 5 °C, 17 °C, and 37 °C for up to 6 h. At 0.5, 2, and 6 h of storage, sperm motility, viability, mitochondrial membrane potential, and intracellular reactive oxygen species (ROS) levels and apoptotic changes were measured. Antioxidant and oxidant levels were tested in boar sperm (SPZ) and their surrounding environment (SN) at each timepoint. The results showed significant effects of post-thaw storage time and temperature and an impact on boar sperm quality (total and progressive motility, VCL, viability, acrosome integrity), early and late sperm apoptotic changes, and changes in MDA levels in SPZ and SN. Compared to storage at 5 °C and 37 °C, frozen-thawed semen samples stored at 17 °C displayed better sperm quality, less apoptotic levels, and lower levels of SPZ MDA and SN MDA. Notably, post-thaw storage at 17 °C extended boar sperm lifespan up to 6 h without obvious reduction in sperm quality. In conclusion, storage of frozen-thawed boar semen at 17 °C preserves sperm quality for up to 6 h, which facilitates the use of cryopreserved boar semen for field artificial insemination.

## 1. Introduction

Cryopreservation allows for the long-term storage and long-distance transport of boar semen and provides flexibility for artificial insemination (AI), regardless of time and space limitations [1]. In addition, the cryopreserved boar semen could be a backup when infectious diseases break out because it permits timely pathogen detection before AI and therefore, regular production on farms. The use of frozen-thawed boar semen for porcine production is evaluated by producers, taking both technical and economic considerations into account. To date, studies have reported that the post-thaw recovery (sperm survival) rate of boar sperm varies from 25% to 60%, with which single or multiple inseminations of sows or gilts obtains a conception rate of 70–85% and a litter size of 11–12 piglets [2]. The fertility results of cryopreserved boar sperm are still lower than those of fresh semen (14 piglets per litter) [2] due to the alterations in sperm motility, membrane status, mitochondrial functionality, and DNA integrity induced by the freezing and thawing procedure [1]. This could be the reason that fertility results are difficult to compensate by literally increasing the number of frozen-thawed sperm that are inseminated. As a result, cryopreserved boar semen is not widely accepted in commercial pig farms but is primarily utilized for export or the conservation of the valuable genes of sires.

Researchers are seeking new approaches to overcome the challenge of improving the quality and fertility of cryopreserved boar sperm. Due to the variations in sperm freezability between individual boars or ejaculates [3], extensive work has been conducted to identify certain biomarkers for boar ejaculate selection before cryopreservation. Antifreeze protein type Ⅲ is related to intracellular ice crystal formation during the freezing process and has been added to freezing extender [4]. It has been demonstrated that miRNA in small extracellular vesicles present in both sperm and seminal plasma is associated with boar sperm cryotolerance [5]. It has been suggested that sperm proteins, e.g., aquaporins [6] and acrosin [7], are good biomarkers for boar semen freezability. Meanwhile, proteins [8], antioxidants [9], and cytokines [10] in seminal plasma are well-correlated with post-thaw boar sperm survival, which could be another type of biomarker for predicting semen cryotolerance. Another big challenge to deal with is the oxidative stress (OS) induced by the overproduction of reactive oxygen species (ROS) during the freezing and thawing process. Chemicals (myoinositol [11], dithiothreitol and glutathione [12], and quercetin [13]) or compounds (palm kernel meal protein hydrolysates [14], natural honey [15], and lipoproteins [16]) with antioxidant properties have been added to freezing or thawing extender to protect against OS-induced damages in sperm quality and functionality. The supplementation of antioxidants showed positive effects on post-thaw sperm quality by reducing OS and therefore, resulted in improved fertility [11,17]. Apart from the above-mentioned reports, attempts to cryopreserve boar semen in fractions [9] build a fine model to investigate the variations in sperm proteins and seminal plasma components that contribute to sperm cryosurvival [1]. Further optimization of freezing and thawing protocols [3] have been conducted, as well. For instance, the single-layer colloid centrifugation before freezing [18] or post-thaw [19] has been demonstrated to be a useful tool in selecting sperm of highest quality. In the previously mentioned literature, marked progress in boar semen cryopreservation has been achieved. The next step is to improve the application of frozen-thawed semen in the field. There are various factors that influence field results after AI with cryopreserved boar semen. These include sperm selection and holding time before freezing, packaging, temperature curve and cryoprotectants used during freezing, thawing methods and post-thaw sperm quality at thawing, AI techniques used, time intervals between thawing and AI, ovulation estimation, and doses of frozen semen used during the AI process [20]. In this study, we focused on the time interval between thawing and AI, aiming to extend the lifespan of frozen-thawed boar sperm, thus providing time flexibility for AI. Usually, AI is recommended to be performed within 30 min post-thaw [2] because the quality and functionality of boar sperm cannot be maintained for an extended time when it is incubated at 37 °C. However, the distance between where semen is thawed and where the AI is performed demands time flexibility in performing AI and longer temporary storage of post-thaw boar semen. To date, only one study has investigated the effects of post-thaw storage temperature on boar sperm lifespan, indicating that post-thaw storage at 17~26 °C for 2 h facilitates the preparation of frozen-thawed doses [21]. The objective of this study was to evaluate (ⅰ) the effect of post-thaw storage at 5 °C (initial temperature of boar semen after thawing at 37 °C for 20 s), 17 °C (conventional semen storage temperature for liquid semen), and 37 °C (constant storage at thawing temperature) on boar sperm survival and (ⅱ) to discover the underlying molecular changes. The findings will provide strong support for the wide application of AI with frozen boar semen.

## 2. Materials and Methods

### 2.1. Reagents and Media

The chemicals involved in the present study were of analytical grade and purchased from Sigma-Aldrich Co. (Shanghai, China). The extender used for the boar semen was Beltsville thawing solution (BTS), composed of 205.0 mM glucose, 20.4 mM Na_3_C_6_H_5_O_7_, 10.0 mM KCl, 15.0 mM NaHCO_3_, 3.6 mM EDTA, and 0.05 mM kanamycin sulfate (pH 7.2–7.4 and 290–300 mOsmol/kg). Freezing extender Ⅰ contained 80% (v:v) tris-citric acid-glucose extender (111 mM Trizma base, 31.14 mM monohydrate citric acid, 185 mM glucose, 0.05 mM kanamycin sulfate, pH 7.14, and 314 mOsmol/kg) and 20% (v:v) egg yolk (pH 7.2 and 295–300 mOsmol/kg). Freezing extender Ⅱ consisted of freezing extender Ⅰ (89.5%, v:v), glycerol (9%, v:v) and Equex STM (1.5%, v:v, Nova Chemical Sales, Scituate, MA, USA) (pH 6.2 and 1700–1730 mOsmol/kg).

Fluorescent probes propidium iodide (PI, P3566), Mitotracker Deep Red 633 (M22426), and 5-(and-6) chloromethyl-20,70-dichlorodihydrofluorescein diacetate acetyl ester (CM-H_2_DCFDA, C6827) were purchased from Thermofisher (Waltham, MA, USA). Fluorescent probes Hoechst 33,342 (H42, B2261) and fluorescein isothiocyanate-conjugated peanut agglutinin (FITC-PNA, L7381) were purchased from Sigma Aldrich (Shanghai, China).

### 2.2. Semen Samples and Cryopreservation

Boar semen ejaculates used in this study were provided by a commercial AI station (Henan Swinegenes Co., Ltd., Hebi, China;). The five Landrace boars (1~2 years old) used in this study were engaged in conventional fresh semen dose production. The boars involved were healthy, mature, fertile, raised in environmentally controlled conditions, and fed with commercial feed according to semen donor requirements. A sperm-rich fraction of the ejaculate was collected using the gloved-hand method. In total, five ejaculates (one ejaculate per boar) were collected for this study. After semen collection, the ejaculates were immediately diluted 1:1 (v:v) with the BTS extender. Thereafter, the seminal quality was evaluated, and only samples that met quality requirements in terms of sperm motility > 70%, sperm viability > 70%, and abnormality < 15% were used for cryopreservation.

The five semen samples obtained were cryopreserved following a 0.5-mL straw freezing procedure reported by Li et al. (2018) with slight modifications [9]. Briefly, the samples were centrifuged at 2400× *g* for 3 min (5810R, Eppendorf, Shanghai, China) after a quality examination. The harvested sperm pellets were resuspended in freezing extender Ⅰ to a concentration of 1.5 × 10^9^ sperm/mL and cooled to 5 °C for 150 min. Thereafter, sperm samples were reextended with freezing extender Ⅱ to a final concentration of 1.0 × 10^9^ sperm/mL. Immediately, the extended sperm samples were packed automatically into 0.5 mL polyvinyl chloride (PVC) French straws (Minitüb, Tiefenbach, Germany) and frozen using a programmed bio-freezer (IceCube 14M, Minitüb, Tiefenbach, Germany). The program of the bio-freezer was set to: 6 °C/min from 5 °C to −5 °C, 40 °C/min from −5 °C to −80 °C, 30 s of equilibrium at −80 °C, and 70 °C/min from −80 °C to −140 °C. Afterward, the straws were plunged into liquid nitrogen and stored in a liquid nitrogen container for more than 1 week. Straws randomly chosen were thawed at 37 °C for 20 s in a circulating water bath and extended with BTS (1:1, v:v). Three straws from each boar were thawed. For each replicate, a pooled semen sample was made by mixing thawed semen from two boars. In total, three replicates were performed. The pooled semen samples were aliquoted into three parts and stored at different temperatures (5 °C, 17 °C, 37 °C) in the dark for 6 h. At 0.5 h, 2 h, and 6 h of post-thaw storage, one part of the semen samples was taken out for sperm quality and functionality assays while another part was centrifuged (2400× *g*, 3 min) to isolate sperm (SPZ) from the surrounding fluid environment (SN). The yielded sperm samples were washed with PBS by centrifuging them three times (2400× *g*, 3 min). The recovered SN was centrifuged (2400× *g*, 3 min) three times, and the absence of sperm was confirmed under microscopy. The obtained SPZ and SN samples were stored at −80 °C for antioxidant and oxidant assays.

### 2.3. Post-Thaw Measurements of Sperm Quality and Functionality

Sperm motility was evaluated using a computer-assisted sperm analysis system (CASA, ISASV1^®^; Proiser R + D, Paterna, Spain). A total of 25 frames were recorded per second, the particle area was set between 10 and 80 μm^2^, and the velocity was set at 20 μm/s < slow < 25 μm/s <medium < 45 μm/s <rapid. Briefly, frozen-thawed semen samples were extended with BTS to a final concentration of 25 × 10^6^ sperm/mL. A total of 5 µL was taken out of each semen sample and placed in a Makler counting chamber (Sefi Medical Instruments, Haifa, Israel) preheated to 38 °C. A minimum of 400 sperm from four or five fields were captured for analysis. Total sperm motility (TM) was recorded as percentage of total motile spermatozoa displaying an average path velocity ≥ 20 µm/s. Progressive sperm motility (PM) was recorded as the percentage of motile spermatozoa showing rapid and progressive movement (straight line velocity ≥ 40 µm/s and straightness of the average path > 45%). Sperm velocity parameters in terms of curvilinear velocity (VCL, µm/s), straight-line velocity (VSL, µm/s), and average path velocity (VAP, µm/s); sperm linearity parameters, including linearity of sperm movement (LIN, %), straightness of average path (STR, %), and wobble coefficient (WOB, %); and sperm vigor parameters in terms of amplitude of lateral head displacement (ALH, µm) and beat cross frequency (BCF, Hz) were recorded.

A flow cytometer (CytoFLEX S, Beckman Coulter Inc., Brea, CA, USA) was employed for the analysis of sperm viability, mitochondrial membrane potential, intracellular ROS production, and sperm apoptotic changes.

The sperm viability in terms of plasma and acrosomal membrane integrity was measured using a triple-fluorescence staining method [22]. Briefly, a 100 μL sperm sample (25 × 10^6^ sperm/mL) was added to a cytometric tube containing 3 μL H-42 (0.05 mg/mL in PBS), 2 μL PI (0.5 mg/mL in PBS), and 2 μL PNA-FITC (200 μg/mL in PBS) and incubated at 37 °C in the dark for 10 min. Before uploading semen samples to flow cytometry, an addition of 400 μL PBS was performed to adjust the sperm concentration to 5 × 10^6^ sperm/mL. The percentage of spermatozoa showing negative PI and negative PNA-FITC was recorded as sperm viability. The sperm population exhibiting negative PI and positive PNA-FITC was recorded as viable spermatozoa with damaged acrosome membranes.

The mitochondrial membrane potential was determined using Mitotracker Deep Red 633 staining following a procedure described by Li et al. (2023) [23]. Briefly, cytometric tubes containing a 100 μL sperm sample (25 × 10^6^ sperm/mL) was loaded with 3 μL of H-42 (0.05 mg/mL in PBS), 2 μL of PI (0.5 mg/mL in PBS), and 5 μL of Mitotracker (0.2 μM of PBS in a stock solution of 1 mM in DMSO) and incubated at 37 °C for 15 min. Before analysis, 400 μL of PBS was added to the cytometric tube to extend the semen samples to a concentration of 5 × 10^6^ sperm/mL. The sperm population exhibiting negative PI and positive Mitotracker were considered viable sperm with high mitochondrial membrane potential.

The intracellular levels of ROS were determined using CM-H2DCFDA staining following the procedure described by Guthrie and Welch [24]. Briefly, a 50 μL semen sample (25 × 10^6^ sperm/mL) was stained with 1.5 μL of H-42 (0.05 mg/mL in PBS), 1 μL of PI (0.5 mg/mL in PBS), and 1 μL of CM-H2DCFDA (1 mM in DMSO). An additional 1 μL of TBH (458139, Sigma, Shanghai, China 70% in distilled water) was added to the positive control group. For each cytometric tube, 950 μL of PBS was added. The samples were thereafter incubated at 37 °C for 30 min in the dark before analysis. The results were presented as the fluorescence intensity of 1 million viable sperm cells showing negative PI and positive DCF.

Sperm apoptotic levels were assessed using a commercial kit (A211-02, Vazyme, China). Briefly, 20 µL semen samples (25 × 10^6^ sperm/mL) were taken and centrifuged (300× *g*, 5 min, 4 °C) to recover sperm samples. The obtained sperm samples were centrifuged (300× *g*, 5 min, 4 °C) and washed twice with precooled PBS and reextended with 100 µL of Annexin-binding buffer. Thereafter, sperm samples were stained with 5 µL Annexin V-FITC (original solution) and 5 µL PI (0.5 mg/mL in PBS) and incubated in the dark at room temperature (20–25 °C) for 10 min. Before analysis, 400 µL of Annexin-binding buffer was loaded into the cytometric tube to adjust the sperm concentration to 1 × 10^6^ sperm/mL. The percentages of viable sperm with negative PI and positive Annexin V were recorded as early apoptotic level. The percentages of sperm with positive PI and positive Annexin V were recorded as late apoptotic level.

### 2.4. Measurements of Antioxidants and Oxidants in SPZ and SN

A microplate reader (PowerWave XS; Bio-Tek Instruments, Winooski, VT, USA) was utilized to determine the contents or activities of antioxidants and oxidants in boar SPZ and SN, as previously described by Li et al. (2023) [23]. SN samples were measured directly. For SPZ samples, a total of 25 × 10^6^ cells were recovered by centrifugation (1600× *g*, 5 min) and subsequently re-diluted with 360 μL of 1% Triton X-100 to lyse cells at 4 °C for 20 min. Thereafter, the supernatant was harvested by centrifugation (4000× *g*, 30 min, 4 °C) and kept on ice until use.

The malondialdehyde (MDA) contents indicating lipid peroxidation level were assayed using a commercial kit (A003-1-2, Nanjing Jiancheng, China), following the manufacturer’s instructions. The principle of the assay is that the MDA-TBA adduct generated from the reaction of MDA in the sample with thiobarbituric acid (TBA) can be quantified colorimetrically (OD = 532 nm). The content of MDA was expressed in nM/mL. The total oxidant status (TOS) representing small molecules with oxidant properties was measured following the procedure described by Erel [25]. The oxidation reaction between oxidants in the sample and the ferrous ion–chelator complex results in the generation of ferric ion, which can be prolonged using glycerol. Thus, the ferric ion is abundant in the reaction medium. In an acidic medium, the ferric ion and chromogen form a colored complex, which is related to the total amount of oxidant molecules present in the sample. The intensity of color can be measured spectrophotometrically. The results were presented in terms of micromolar hydrogen peroxide equivalent per liter (μM H_2_O_2_ Equiv./L). A commercial kit (S0121, Beyotime, Shanghai, China) was used to determine the total antioxidant capacity (TAC) in terms of the Trolox-equivalent antioxidant capacity. The principle of the assay is based on the decolorization of 2,2′-azinobis-(3-ethylbenzothiazoline-6-sulfonate) when it reacts with antioxidants. The color change is detected at 414 nm using a microplate reader. The results of the TAC assay were expressed in mM/L.

Superoxide dismutase (SOD)-like activity was measured using a commercial kit (A001-3-2, Nanjing Jiancheng, Nanjing, China). The action of xanthine oxidase produces superoxide anions. The SOD-like antioxidants can catalyze the dismutation of the superoxide anions into hydrogen peroxide and oxygen. The assay is based on that superoxide anions act on WST-1 to produce a water-soluble formazan dye that can be detected by an increase in absorbance at 450 nm. Thus, the amount of formazan produced is related to the SOD-like activity in the sample. A 50% SOD-like antioxidant inhibition corresponds to the SOD-like activity of 1 U. The SOD-like activity was calculated using the equation:$$SOD–like\;antioxidants\;inhibition\;\left(\%\right)=\frac{\left(A\;blank1-A\;blank3)-(A\;sample-A\;blank2\right)}{\left(A\;blank1-A\;blank3\right)}\;SOD–like\;activity\left(U/mL\right)=\frac{SOD–like\;antioxidants\;inhibition}{50\%}\times Dilution\;ratio$$

The glutathione peroxidase 5 (GPX5)-like activity was measured using a commercial kit (ml622032, Shanghai Enzyme Link, Shanghai, China) based on an enzyme-linked immune bi-antibody sandwich two-step method. Samples were placed in wells and pre-coated with GPX5 capture antibody. An HRP-labeled detection antibody was added and washed afterward. TMB was used as a substrate for color development. The color change in the TMB catalyzed using GPX5, which can be detected at 450 nm using a microplate reader, corresponds to the GPX5-like activity in samples. The results were expressed as U/mL.

### 2.5. Statistical Analysis

The software IBM SPSS (version 20.0) was employed to analyze the data in this study. The normality of the data was analyzed using the Kolmogorov–Smirnov test based on the residuals. General linear models were utilized to test the interactive effect between storage time and temperature on sperm quality and functionality, as well as antioxidant and oxidant parameters in sperm and their surrounding environment. The multivariant one-way ANOVA was utilized to perform comparisons between post-thaw storage time and temperature groups, followed by LSD multiple-comparison tests. The data were presented as the means ± SEMs. A *p* value < 0.05 was set as a statistically significant threshold.

## 3. Results

### 3.1. Effect of Post-Thaw Storage on Boar Sperm Motion Parameters

General linear analysis showed that there was an interaction (*p* < 0.05) between post-thaw storage time and storage temperature affecting total sperm motility (TM) and progressive sperm motility (PM) (Figure 1). After post-thaw storage for 0.5 h, semen samples stored at 5 °C showed lower (*p* < 0.05) values of TM than that at 37 °C. After 2 h of post-thaw storage, semen samples stored at 17 °C exhibited higher (*p* < 0.05) values of TM than those stored at 5 °C and 37 °C. After 6 h of post-thaw storage, semen samples stored at 17 °C showed higher (*p* < 0.05) values of TM than those stored at 37 °C. Post-thaw storage at 5 °C and 17 °C maintained sperm TM during a prolonged storage time, whereas a decline (*p* < 0.05) of TM with post-thaw storage time was observed when semen samples were stored at 37 °C. The values of PM were influenced by storage temperature at 2 h of post-thaw storage; PM was higher (*p* < 0.05) when stored at 17 °C than it was when stored at 5 °C. After 6 h of post-thaw storage, a tendency for a decrease (*p* = 0.064) in PM was observed when semen samples were stored at 37 °C in comparison with those stored at 17 °C. Extended storage at 37 °C resulted in damage (*p* < 0.05) to PM. Storage at 5 °C and 17 °C maintained sperm PM.

General linear analysis of sperm kinetic parameters showed that an interaction (*p* < 0.05) between post-thaw storage time and storage temperature was only found in VCL (Figure 2 and Figure 3). Because no interaction between post-thaw storage time and storage temperature was found in the rest of the kinetic parameters, the data were combined to analyze the effect of storage time and temperature, respectively. As is shown in Figure 2, no influence of storage temperatures on VCL was observed after post-thaw storage for 0.5 h. After 2 h of storage, semen samples stored at 17 °C displayed higher (*p* < 0.05) values of VCL than those stored at 5 °C and 37 °C. The values of VCL were lower (*p* < 0.05) in samples stored at 37 °C than in those stored at 5 °C. After 6 h of storage, values of VCL in samples stored at 17 °C were similar to those stored at 5 °C, being higher (*p* < 0.05) than those stored at 37 °C. Extended storage time decreased VCL values when semen samples were stored at 5 °C (*p* = 0.055) and 37 °C (*p* < 0.05). Extended storage at 17 °C maintained sperm VCL. Prolonged post-thaw storage time led to a decrease in sperm velocity in terms of VSL (*p* < 0.05) and VAP (*p* < 0.05) (Figure 2) and a decrease in sperm vigor in terms of ALH (*p* < 0.05) and BCF (*p* < 0.05) (Figure 3). Storage at 17 °C showed similar VAP values to samples stored at 5 °C and better VAP values (*p* < 0.05) than those stored at 37 °C (Figure 2). Increased storage temperature resulted in higher values in sperm linearity in terms of LIN (*p* < 0.05) and STR (*p* < 0.05) (Figure 2 and Figure 3). Semen samples stored at 17 °C showed similar sperm vigor in terms of ALH to those stored at 5 °C and higher ALH (*p* < 0.05) than those stored at 37 °C (Figure 3). No influence of post-thaw storage temperature on sperm VSL, WOB, or BCF was observed.

### 3.2. Effect of Post-Thaw Storage on Boar Sperm Morphological Status

General linear analysis showed that there was an interaction (*p* < 0.05) between post-thaw storage time and storage temperature affecting sperm viability and the percentage of damaged acrosome membrane. After 6 h of post-thaw storage, semen samples stored at 17 °C showed better (*p* < 0.05) sperm viability than those stored at 5 °C and 37 °C. Extended storage at 37 °C induced a decline (*p* < 0.05) in sperm viability. Accordingly, semen samples stored at 17 °C exhibited lower (*p* < 0.05) levels of acrosome damage than those stored at 5 °C and 37 °C. Storage at 17 °C maintained acrosome membrane integrity during extended storage time well, whereas prolonged storage time negatively influenced acrosome membrane integrity when semen was stored at 5 °C (*p* < 0.05) and 37 °C (*p* < 0.05). No effects of either post-thaw storage time or storage temperature on mitochondrial membrane potential were found. See the results in Figure 4.

### 3.3. Effect of Post-Thaw Storage on Boar Sperm Apoptotic Changes

General linear analysis showed that there was an interaction (*p* < 0.05) between post-thaw storage time and storage temperature affecting sperm apoptotic levels (Figure 5). No influence of storage temperature on early and late apoptotic levels of sperm stored for 0.5 h after thawing was observed. The elevated storage temperature increased early and late apoptotic levels of sperm stored for 2 h and 6 h; the apoptotic levels were higher (*p* < 0.05) when semen samples were stored at 37 °C. A prolonged storage time led to an increase (*p* < 0.05) in sperm early apoptosis when semen samples were stored at 5 °C, 17 °C, and 37 °C, whereas an increase (*p* < 0.05) in sperm late apoptosis was only found when semen samples were stored at 37 °C.

### 3.4. Effect of Post-Thaw Storage on Antioxidant and Oxidant Levels in Boar Sperm and Their Surrounding Environment

An interaction (*p* < 0.05) between post-thaw storage time and temperature affecting antioxidant and oxidant parameters was only observed in SPZ MDA and SN MDA. The data for the rest of antioxidant and oxidant parameters were combined to analyze the effect of storage time and temperature. As is shown in Figure 6, elevated post-thaw storage temperatures (17 °C and 37 °C) resulted in lower (*p* < 0.05) levels of SPZ MDA and SN MDA than those stored at 5 °C when semen samples were stored for 0.5 h and 2 h. In contrast, when storage time reached 6 h, semen samples stored at 37 °C exhibited higher (*p* < 0.05) levels of SPZ MDA than those stored at 5 °C and 17 °C; these samples reached the highest (*p* < 0.05) levels of SN MDA at 37 °C and lowest (*p* < 0.05) levels of SN MDA at 17 °C. The levels of SPZ MDA and SN MDA increased (*p* < 0.05) with storage time. Post-thaw storage at 17 °C showed the most positive effects in terms of SPZ MDA and SN MDA levels. No apparent influences of post-thaw storage time and temperature on sperm intracellular ROS levels were observed.

As is displayed in Figure 7, post-thaw storage time had no effect on SPZ TOS levels. A decrease (*p* < 0.05) in SN TOS levels was observed at 2 h of storage compared with those at 0.5 h of storage. No effects of post-thaw storage temperature on SPZ TOS and SN TOS levels were observed. No influences of post-thaw storage temperatures and time on SPZ TAC and SN TAC levels were found.

As is shown in Figure 8, no impact of post-thaw storage time and temperatures on SPZ SOD and SPZ GPX5 levels were observed. Prolonged storage time increased (*p* < 0.05) SN SOD and SN GPX5 levels. No influences of post-thaw storage temperatures on SN SOD and SN GPX5 levels were found.

## 4. Discussion

Extensive work has been conducted to improve the quality and fertility of frozen-thawed boar semen. The screening of cryoprotectants, additives in freezing and thawing extender, and novel methods for optimization of cryopreservation protocols have resulted in significant progress in post-thaw sperm quality and reproductive performance in the field, which is relatively comparable to that of fresh semen [2]. Still, there are obstacles in the on-farm application of frozen semen, among which the short lifespan of post-thaw boar sperm is a major issue. This study aimed to test lower post-thaw storage temperature on boar sperm lifespan. The results indicated that storage at 17 °C obviously extended the lifespan of frozen-thawed boar sperm, owing to lower levels of sperm apoptotic changes and less OS induced in comparison with samples stored at 5 °C and 37 °C. Post-thaw storage at 17 °C maintained boar sperm quality up to 6 h, providing sufficient time for the preparation of frozen-thawed semen doses and their transportation for AI.

Sperm motion parameters are vital for the evaluation of frozen-thawed boar semen because they are the variants most influenced by cryostorage [26]. Sperm kinetic parameters are susceptible to the freezing procedure, especially after the removal of seminal plasma using centrifugation at 17 °C and during cooling at 5 °C, during which sperm hyperactivated movement increases [27]. This study demonstrated a significant effect on post-thaw storage time and temperature and an impact on boar sperm movement patterns. Compared to 5 °C and 37 °C, storage at 17 °C exhibited higher total and progressive motility and kinetic parameters in terms of VCL, VAP, ALH, and BCF but lower linearity in terms of LIN and STR. In addition, post-thaw storage at 17 °C maintained boar sperm motion parameters well. Our findings indicated the advantages of post-thaw storage at 17 °C in conserving sperm movement status over that at 5 °C or 37 °C. Post-thaw storage at temperatures that are too low or too high exerts a negative effect on sperm quality, which can be explained by the report that storage of frozen-thawed boar sperm at 17 °C or 26 °C for 2 h has no influence on sperm motility [21]. The favorable results of sperm motion parameters manifested at 17 °C of storage will promote the practical application of cryopreserved boar semen for AI as sperm total and progressive motility, and individual kinetic parameters like velocity and vigor are associated with farrowing rate and total number of born piglets [28,29].

In contrast, sperm membrane status remains relatively stable during liquid or frozen storage. The extender type and components may impact sperm viability either in liquid [30] or frozen form [31]. Furthermore, thawing extender and post-thaw storage time influence boar sperm membrane integrity. Although sperm viability is held for 6 h at 17 °C or 26 °C, a dramatic decline occurs after 24 h. A higher storage temperature (37 °C) induces a continuous decrease in sperm viability [21]. A similar result was observed in the present study. Post-thaw storage at 5 °C and 17 °C preserved sperm plasmatic and acrosomal membrane integrity, whereas apparent damage with time was found at 37 °C. Notably, storage at 17 °C maintained sperm membrane integrity well for up to 6 h. It is suggested that the mitochondrial membrane potential damage takes place prior to or concurrently with sperm motility loss [32]. However, no significant effect of post-thaw storage time and temperature was observed on sperm mitochondrial membrane potential in the present study. The loss in sperm motility may be attributed to an appropriate pH environment [33] or the storage temperature, as evidenced [34].

Temperature management is crucial for boar sperm preservation because they are sensitive to temperature changes [35] ascribed to the rich abundance of unsaturated fatty acids in the plasma membrane [36]. Lipid architecture and the fluidity of the sperm plasma membrane can be altered by temperature changes [37,38], resulting in damage to sperm quality and functionality. Therefore, storage temperature at 17–25 °C is recommended for boar semen [39] because sperm quality loss is observed at >20 °C or <15 °C [40]. The results of this study indicate that post-thaw storage at 17 °C leads to better sperm quality than that at 5 °C and 37 °C. Accordingly, lower levels of apoptotic changes in boar sperm were observed at 17 °C in comparison with those at 5 °C and 37 °C, which could explain the results. Our previous study found that temperature elevation promotes boar sperm apoptotic changes [22]. Similarly, post-thaw storage at 37 °C showed obviously higher levels of early and late sperm apoptosis than that at 5 °C and 17 °C in this study, which is consistent with the previous report.

Antioxidant and oxidant levels in boar sperm and the surrounding fluid have been demonstrated to be associated with boar sperm quality and functionality during liquid storage through influencing the phosphorylation of AMP-activated protein kinase (AMPK) [23]. In the present study, post-thaw storage at 17 °C resulted in the lowest SPZ MDA and SN MDA, although their levels at all the storage temperatures increased considerably with time, especially at 37 °C. The apoptotic changes in boar sperm could be the result of increased MDA, an indicator of lipid peroxidation, induced by higher post-thaw storage temperature, as was observed in this study. The occurrence of lipid peroxidation may further induce ROS production, promoting the apoptotic response in boar sperm [41]. As a result, sperm motility and membrane integrity damage were observed. The SN TOS levels decreased with post-thaw storage time, whereas the levels of SN SOD and SN GPX5 increased, which indicates the dynamic response of sperm to OS during post-thaw storage. In addition, the SN SOD and SN GPX5 have been demonstrated to be involved in boar sperm resistance to cryopreservation [9]. Supplementation with astaxanthin, a substance with antioxidant properties, in thawing extender decreases the apoptotic levels in frozen-thawed boar sperm [42], which implies the positive role of antioxidants in boar sperm resistance to post-thaw storage. Boar sperm may respond to post-thaw-induced OS by promoting AMPK phosphorylation [43] and decreasing apoptotic levels through both the death receptor- and mitochondria-mediated apoptotic pathways [44], as evidenced by supplementation of resveratrol during freezing.

## 5. Conclusions

Post-thaw storage of boar semen at 17 °C showed the best effect on sperm quality maintenance with the least induction of sperm apoptosis and oxidative stress. Storage of frozen-thawed boar sperm at 17 °C for up to 6 h provides sufficient time for semen dose preparation and transportation for field AI practice. Thus, the results of the present study enhance further research regarding in vivo experiments to corroborate the use of frozen-thawed boar semen stored at 17 °C for 6 h post-thaw for AI in field conditions.

## Figures and Tables

**Figure 1 animals-14-00087-f001:**
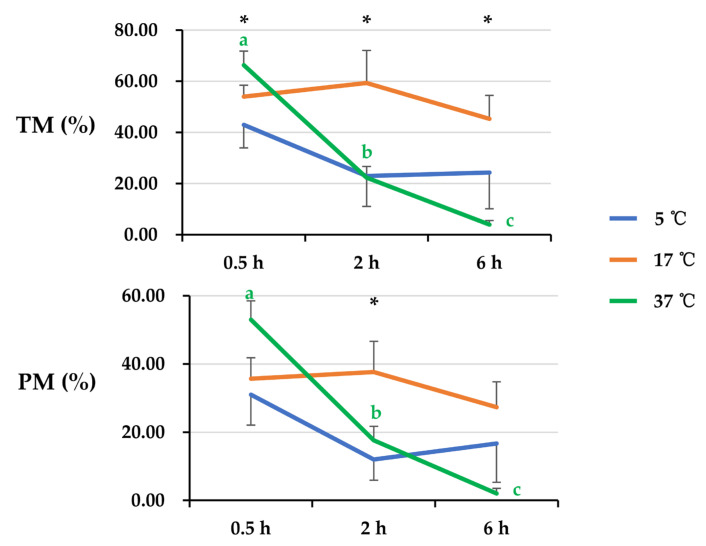
The line graph shows total sperm motility (TM) and progressive sperm motility (PM) changes of frozen-thawed boar semen affected by post-thaw storage time and temperature. General linear analysis showed interaction between post-thaw storage time and temperature affecting TM and PM. Values of TM and PM were compared among different storage times and temperatures. Data are expressed as the mean ± SEM. The letters a, b, and c denote significant differences between storage times at each storage temperature, *p* < 0.05. * Indicates differences among storage temperatures at each storage timepoint, *p* < 0.05.

**Figure 2 animals-14-00087-f002:**
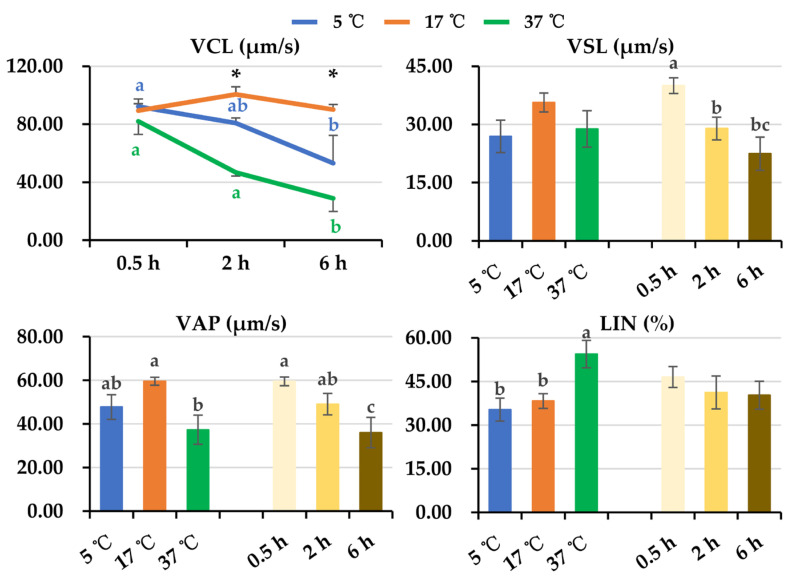
The line graph shows changes in sperm velocity parameter VCL of frozen-thawed boar semen affected by post-thaw storage time and temperature. General linear analysis showed interaction between post-thaw storage time and temperature affecting VCL. Histograms show changes in sperm velocity parameters (VSL and VAP) and linearity parameters (LIN). Because no interaction between post-thaw storage time and temperature was observed in these parameters, the data were combined to analyze the effect of storage time. VCL: curvilinear velocity (µm/s), VSL: straight-line velocity (µm/s), VAP: average path velocity (µm/s), LIN: linearity of sperm movement (%). Data are expressed as the mean ± SEM. The letters a, b, and c denote significant differences between storage temperatures at each storage timepoint or between storage timepoints, *p* < 0.05. * Indicates differences among storage temperatures at each storage timepoint in line graph, *p* < 0.05.

**Figure 3 animals-14-00087-f003:**
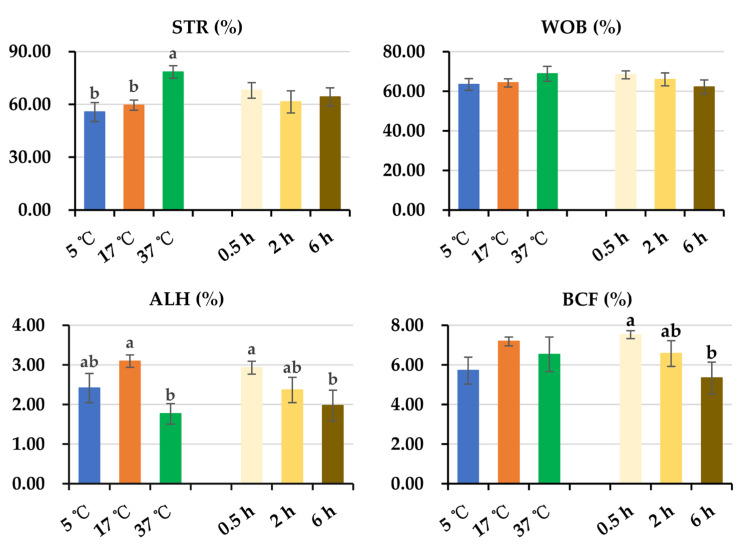
Histograms show changes in sperm linearity parameters (STR, WOB) and vigor parameters (ALH, BCF) during post-thaw storage. Because no interaction between post-thaw storage time and temperature was observed in these parameters, the data were combined to analyze the effect of storage temperature. STR: straightness of the average path (%), WOB: wobble coefficient (%), ALH: amplitude of lateral head displacement (µm), BCF: beat cross frequency (Hz). Data are expressed as the mean ± SEM. The letters a and b denote significant differences between storage temperatures or timepoints, *p* < 0.05.

**Figure 4 animals-14-00087-f004:**
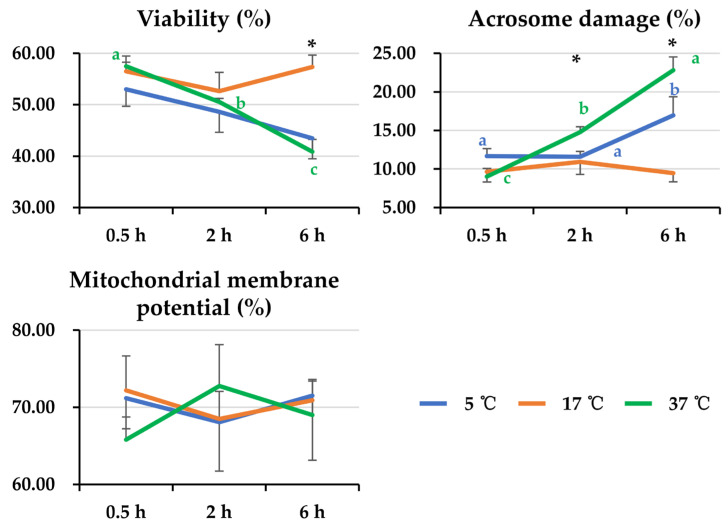
The line graphs show changes in sperm viability, percentage of acrosome membrane damage, and mitochondrial membrane potential of frozen-thawed boar semen affected by post-thaw storage time and temperature. General linear analysis showed interaction between post-thaw storage time and temperature in sperm viability and percentage of acrosome membrane damage but not in sperm mitochondrial membrane potential. Data are expressed as the mean ± SEM. The letters a, b, and c denote significant differences between storage timepoints at each storage temperature, *p* < 0.05. * Indicates differences among storage temperatures at each storage timepoint, *p* < 0.05.

**Figure 5 animals-14-00087-f005:**
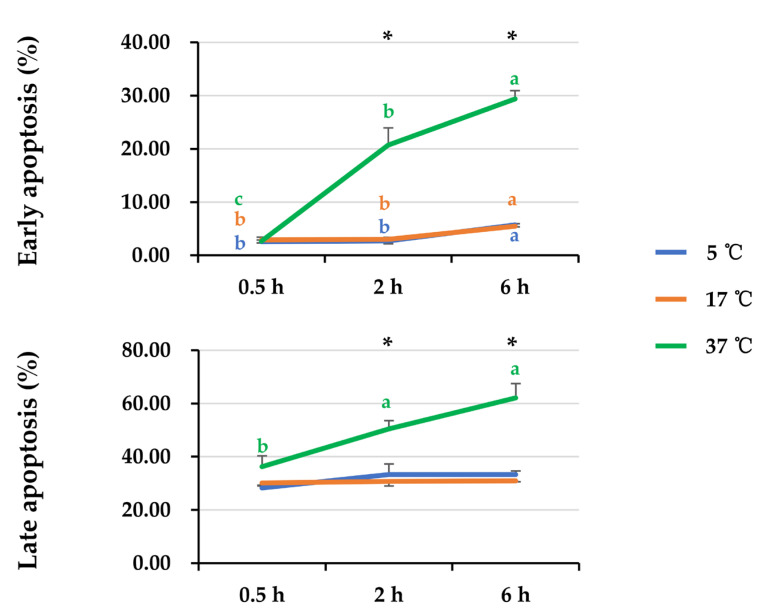
The line graphs show the changes in early and late sperm apoptosis of frozen-thawed boar semen affected by post-thaw storage time and temperature. General linear analysis showed interaction between post-thaw storage time and temperature affecting both early and late sperm apoptosis. Data are expressed as the mean ± SEM. The letters a, b, and c denote significant differences between storage timepoints at each storage temperature, *p* < 0.05. * Indicates differences among storage temperatures at each storage timepoint, *p* < 0.05.

**Figure 6 animals-14-00087-f006:**
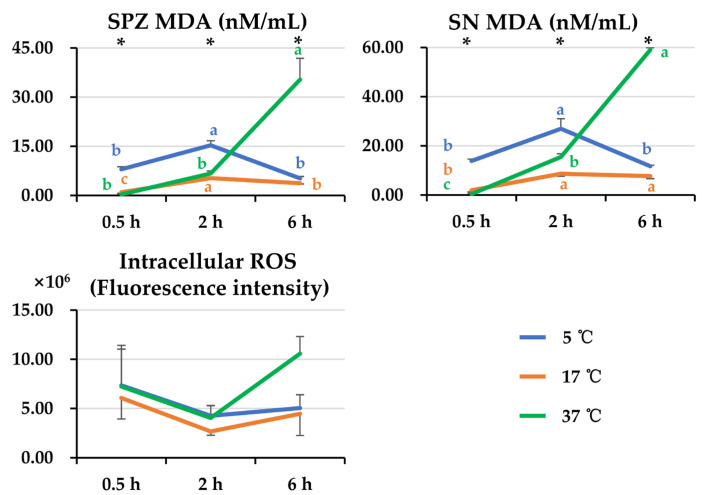
The line graphs show the changes in SPZ MDA, SN MDA, and intracellular ROS levels of frozen-thawed boar semen affected by post-thaw storage time and temperature. General linear analysis showed interaction between post-thaw storage time and temperature affecting SPZ MDA and SN MDA but not intracellular ROS levels. SPZ: frozen-thawed boar sperm, SN: the surrounding environment of frozen-thawed boar sperm, MDA: malondialchehyche. Data are expressed as the mean ± SEM. The letters a, b, and c denote significant differences between storage timepoints at each storage temperature, *p* < 0.05. * Indicates differences among storage temperatures at each storage timepoint, *p* < 0.05.

**Figure 7 animals-14-00087-f007:**
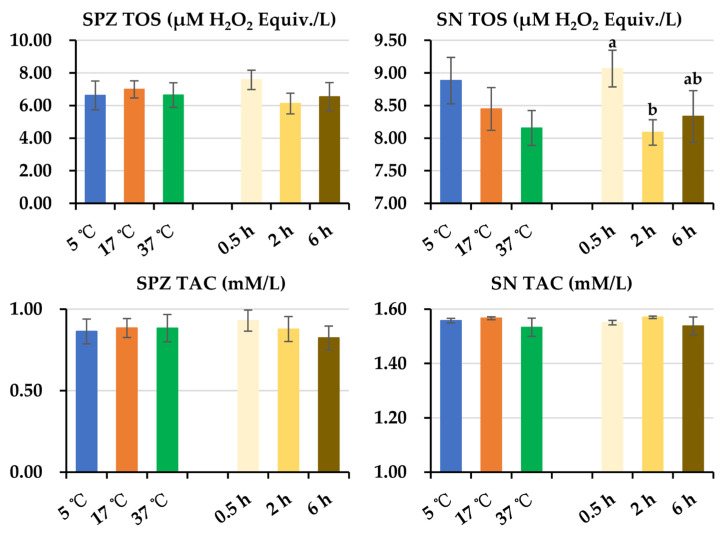
Histograms show the changes in SPZ TOS, SN TOS, SPZ TAC, and SN TAC levels of frozen-thawed boar semen. Because no interaction between post-thaw storage time and temperature was observed in both parameters, the data were combined to analyze the effect of storage time and temperature, respectively. SPZ: frozen-thawed boar sperm, SN: the surrounding environment of frozen-thawed boar sperm, TOS: total oxidant status, TAC: total antioxidant capacity. Data are expressed as the mean ± SEM. The letters a and b denote significant differences between storage timepoints, *p* < 0.05.

**Figure 8 animals-14-00087-f008:**
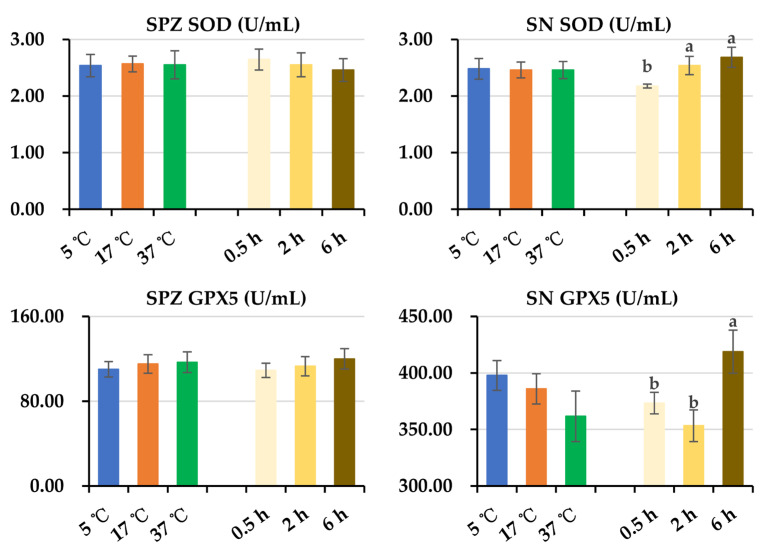
Histograms show the changes in SPZ SOD, SN SOD, SPZ GPX5, and SN GPX5 levels of frozen-thawed boar semen. Because no interaction between post-thaw storage time and temperature was observed in these parameters, the data were combined to analyze the effect of storage time and temperature, respectively. SPZ: frozen-thawed boar sperm, SN: the surrounding environment of frozen-thawed boar sperm, SOD: superoxide dismutase, GPX5: glutathione peroxidase 5. Data are expressed as the mean ± SEM. The letters a and b denote significant differences between storage timepoints, *p* < 0.05.

## Data Availability

The data presented in this study are available in the study.

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
