# Peer review of "Post-Thaw Storage Temperature Influenced Boar Sperm Quality and Lifespan through Apoptosis and Lipid Peroxidation"

_animals, 2023, doi:10.3390/ani14010087_

Round 1

Reviewer 1 Report

Comments and Suggestions for Authors

The scientific content is of interest in the field of cryopreservation. The experimental design is useful for the clinical application in the pig farm.

However, there is one important concern of the Legend of X axis. Please check all the figures in the manuscript that the Legend of X axis must be the "different time points, 0.5, 2 h, 6h" and the 3 lines must represent different temperatures.

Author Response

Comments and Suggestions for Authors

Point 1. The scientific content is of interest in the field of cryopreservation. The experimental design is useful for the clinical application in the pig farm.

However, there is one important concern of the Legend of X axis. Please check all the figures in the manuscript that the Legend of X axis must be the "different time points, 0.5, 2 h, 6h" and the 3 lines must represent different temperatures.

Response 1: The authors appreciate your valuable suggestions. All the line graphs involved have been modified according to your comments.

Reviewer 2 Report

Comments and Suggestions for Authors

Review Reports

Journal: Animals

Manuscript No: animals-2761668

Post-thaw storage temperature influenced boar sperm quality and lifespan through apoptosis and lipid peroxidation

In general, this manuscript was well-written, the authors try to address the effect of storage temperatures and incubation or storage periods post thawing on semen quality parameters, oxidative stress profiles and apoptotic parameters in semen of boar. Preservation of pig semen at 17°C has higher beneficial than lower or higher temperatures and this is standard procedure and known truth. The research work may be considered after answering the following queries.

Ø  The present study was conducted with use of five ejaculates. Whether this will meet minimum requirement for research? Explain why?

Ø  Write the standard values for CASA parameters

Ø  Figures 1, 2A, 4, 5, and 6 need to be converted in to tables for better understanding, visibility and explanation

Ø  Some of the figures are in colour and some of them are in black and white; place the all the figures either in colour or in black and white for better visibility and understanding of the readers

Ø  Include clear figures of sperm viability, acrosomal integrity, mitochondrion membrane potential, and sperm apoptosis with appropriate marking at different incubation temperatures and periods  

Ø  Include other seminal biochemical profiles such as AST, ALT, LDH, triglycerides, total cholesterol, etc. to assess the effect of temperature and incubation periods on semen quality.

Ø  Further, in-vitro or in-vivo fertility trials need to be conducted to confirm the present findings.

Ø  What about plasma membrane integrity; important test to analyse the membrane integrity; needs to be done

Ø  What about different sperm morphological abnormalities?

Ø  Figures are not clear and not in uniform format. Some figures are with temperatures and time periods and some of them are with only temperature; confusing

Ø  How did you measure osmolarity of the extender?

Author Response

In general, this manuscript was well-written, the authors try to address the effect of storage temperatures and incubation or storage periods post thawing on semen quality parameters, oxidative stress profiles and apoptotic parameters in semen of boar. Preservation of pig semen at 17°C has higher beneficial than lower or higher temperatures and this is standard procedure and known truth. The research work may be considered after answering the following queries.

Point 1. The present study was conducted with use of five ejaculates. Whether this will meet minimum requirement for research? Explain why?

Response 1: Thank you for your valuable comments. Firstly, the five boars involved in this study for semen collection were used in the commercial boar stud for regular production of semen doses for AI. And they were selected by the boar stud for semen cryopreservation due to a good freezability of their semen. Secondly, the aim of this study was to test the effect of the post-thaw temperature on sperm quality. About 60 straws from each ejaculate were made and randomly selected for experiments. In addition, we pooled thawed semen and repeated the experiment three times. In this way, variations in both boar individual and ejaculates were minimized.

Point 2. Write the standard values for CASA parameters

Response 2: Actually, the sperm motion parameters measured by the CASA system vary too much depending on the ejaculates. For the total sperm motility, in China National Standard there is a recommended value, >70%, with which the semen can be used for AI. For the other CASA parameters, there is no recommended values currently.

Point 3. Figures 1, 2A, 4, 5, and 6 need to be converted in to tables for better understanding, visibility and explanation

Response 3: The authors appreciate your suggestion. Those figures have been remade. We set post-thaw storage time as the X axis to compare the effect between different storage temperatures. Besides, those figures you mentioned were used to present the effect of both post-thaw time and temperature, and the interactive effect between the two factors. Therefore, we believe the line graphs work better than tables.

Point 4. Some of the figures are in colour and some of them are in black and white; place the all the figures either in colour or in black and white for better visibility and understanding of the readers

Response 4: Thank you so much for pointing out this problem. The figures have been modified accordingly.

Point 5. Include clear figures of sperm viability, acrosomal integrity, mitochondrion membrane potential, and sperm apoptosis with appropriate marking at different incubation temperatures and periods 

Response 5: The figures have been remade according to your comments.

Point 6. Include other seminal biochemical profiles such as AST, ALT, LDH, triglycerides, total cholesterol, etc. to assess the effect of temperature and incubation periods on semen quality.

Response 6: The authors really appreciate your advice. Actually, we are planning to investigate the effects of storage temperature on sperm metabolism. It could be a interesting study. We expect to disclose how the temperature elevation leads sperm cells to death. However, this study was designed to find a best post-thaw storage method to extend sperm lifespan. For this reason, the metabolism part was not involved.

Point 7. Further, in-vitro or in-vivo fertility trials need to be conducted to confirm the present findings.

Response 7: It is true. This study just provided a practical method to extend lifespan of post-thaw boar sperm. Whether the sperm show good fertility after 6 h of post-thaw storage at 17 ℃ needs to be confirmed. However, during the process of this study and currently, it is very difficult to get oocyte samples for IVF, and sows for AI due to the effect of ASF. 

Point 8. What about plasma membrane integrity; important test to analyse the membrane integrity; needs to be done

Response 8: The evaluation of sperm plasma membrane integrity was previously described in the text. When we evaluate the quality of sperm, motility and membrane integrity are the two most important parameters.

Point 9. What about different sperm morphological abnormalities?

Response 9: The sperm abnormality was evaluated under microscope after semen collection. Only semen samples with abnormality<15% were used for this study.

Point 10. Figures are not clear and not in uniform format. Some figures are with temperatures and time periods and some of them are with only temperature; confusing

Response 10: The figures have been modified.

Point 11. How did you measure osmolarity of the extender?

Response 11: The Ice Point Osmotic Pressure Gauge was used to measure the extender, which is especially working well with freezing extender that contain egg yolk.

Reviewer 3 Report

Comments and Suggestions for Authors

The authors outline the need for improving freeze-thaw protocols for boar semen and also identify a need for a more flexible AI time following thawing, thus, effective post-thaw storage methods are needed.  This study examined the effect of post-thaw storage temperature (5OC , 17OC, or 37OC) on several measures of sperm quality, which were assessed at three different time points following thawing: 0.5 h, 2 h, and 6 h.  In generally, these sperm quality parameters were generally comparable for the three storage temperatures for the 0.5 h time point.  However, sperm samples stored at 17 OC maintained exhibited better sperm quality than those stored at 37OC, and to a lesser extent 5OC , for the 2 h and 6 h time points (especially for the 6-hour time point).   

Overall, the methods are clearly described, and the conclusions of the study match the original aims and objectives.  The provision of the data files as supplementary is appreciated. The results are accurate and well-presented. However, consistency is required in terms of the superscripted letters indicating statistical significance. While the superscript letters make sense in terms of interpreting the figure, sometimes the 'a' superscript is used for the highest value/s (Figures 2b,2d, and 3a) and in others it is the lowest value/s (e.g. Figure 3b). Figures 2a and 1 (TM%) are different again with a being the middle value. There should be a consistent formatting approach for this notation, with 'a' either being the lowest or highest value.  Other than this, I have no major comments. The authors should be commended on this research and well-prepared manuscript.

The manuscript is generally well-written. However, there are a few grammatical errors (or typos) throughout the manuscript.  I have identified some of these below in my minor comments, but a thorough proofread of the manuscript is advised. The references used are appropriate and support the manuscript well. The reference list should be checked carefully for consistent formatting. At present, there are some differences in the format of them. The titles, for example, sometimes have each word starting with a capital letter and others do not. 

For minor comments, please see the PDF of the manuscript with comments and tracked changes.   

Comments on the Quality of English Language

The manuscript is generally well-written. However, there are a few grammatical errors (or typos) throughout the manuscript.  I have identified some of these below in my minor comments, but a thorough proofread of the manuscript is advised. For minor comments/suggestions, please see the PDF of the manuscript with comments and tracked changed

Author Response

The authors outline the need for improving freeze-thaw protocols for boar semen and also identify a need for a more flexible AI time following thawing, thus, effective post-thaw storage methods are needed.  This study examined the effect of post-thaw storage temperature (5OC , 17OC, or 37OC) on several measures of sperm quality, which were assessed at three different time points following thawing: 0.5 h, 2 h, and 6 h.  In generally, these sperm quality parameters were generally comparable for the three storage temperatures for the 0.5 h time point.  However, sperm samples stored at 17 OC maintained exhibited better sperm quality than those stored at 37OC, and to a lesser extent 5OC , for the 2 h and 6 h time points (especially for the 6-hour time point).  

Point 1. Overall, the methods are clearly described, and the conclusions of the study match the original aims and objectives. The provision of the data files as supplementary is appreciated. The results are accurate and well-presented. However, consistency is required in terms of the superscripted letters indicating statistical significance. While the superscript letters make sense in terms of interpreting the figure, sometimes the 'a' superscript is used for the highest value/s (Figures 2b,2d, and 3a) and in others it is the lowest value/s (e.g. Figure 3b). Figures 2a and 1 (TM%) are different again with a being the middle value. There should be a consistent formatting approach for this notation, with 'a' either being the lowest or highest value.  Other than this, I have no major comments. The authors should be commended on this research and well-prepared manuscript.

Response 1: The authors really appreciate your valuable comments. We carefully checked all the figures and made modifications on superscripted letters. The letters used to indicate differences between groups have been adjusted with “a” the highest value.

Point 2. The manuscript is generally well-written. However, there are a few grammatical errors (or typos) throughout the manuscript.  I have identified some of these below in my minor comments, but a thorough proofread of the manuscript is advised. The references used are appropriate and support the manuscript well. The reference list should be checked carefully for consistent formatting. At present, there are some differences in the format of them. The titles, for example, sometimes have each word starting with a capital letter and others do not. For minor comments, please see the PDF of the manuscript with comments and tracked changes.

Response 2: The authors are grateful for your close reading of this manuscript and kind suggestions for revision. We have carefully read your comments on gramma and references format and made revisions accordingly. Your comments are truly helpful for us to improve the quality of this manuscript. Thank you again.

Point 3. How many straws were there in total for each boar? Why not assess those straws separately? (comments in the PDF file)

Response 3: For each boar, one ejaculate was collected for freezing. A total of about 60 straws were made for each ejaculate and randomly selected for thawing. To avoid variations in boars and ejaculates that have been demonstrated by previous studies, this study made pooled semen by mixing straws from two boars.

Reviewer 4 Report

Comments and Suggestions for Authors

The study addresses a important aspect of sperm preservation, seeking to identify the optimal post-thaw storage conditions to extend sperm survival time without compromising quality. While the manuscript presents good findings, there are several questions and concerns that require clarification and further information.

  1. Osmolarity of Extender: Could the authors provide information on the osmolarity of the extender used in the study before the addition of glycerol?
  2. Semen Collection Details: a. How many semen collections were performed per week? b. What was the distance between each semen collection? How raw semen was processed? How was it transported to the lab? What was the temperature of transportation?
  3. Morphological Assessment Protocol: How was the morphology measured as the initial evaluation of semen? Please provide details on the protocol used. Was it expanded morphology?
  4. Dilution Medium for Initial Assessment: When semen was diluted 1:1 with BLT for the initial assessment, did the dilution medium contain glycerol and egg yolk, or was it a basic medium? What was the ratio of dilution and at what temperature?
  5. Concentration of Semen per Straw: What was the concentration of semen per straw used in the study? What was the final volume of semen per dose?
  6. Use of Pooled Semen: Why did the authors choose to use pooled semen from each replicate? Was there a specific reason for not assessing the individual impact of each boar separately? How did you choose the semen per pool?
  7. PBS Composition: a. Was the PBS magnesium-free? b. How about calcium content? c. What was the pH of the PBS?
  8. Sperm Motion Analysis (SCA): During sperm motion analysis in section 2.3, how many frames per second were recorded? Please provide all the settings used for the SCA
  9. Flow Cytometry Assessment: a. What served as the positive control in the flow cytometry assessment? b. Could the authors provide technical information, including wavelengths, excitation and emission details, bandpass information, and longpass information for all individual fluorescent probes used?
  10. Malondialdehyde (MDA) Standard Curve: Regarding the MDA, could the authors provide information on the standard curve used in the study?
  11. Lack of Fertility Data: One major concern is the absence of fertility data in the paper. Is it possible for the authors to include fertility data, such as pregnancy outcomes?
  12. Additional Data Request: a. Is there post-thaw morphology information that could be added to enhance the paper? b. Do the authors have data for individual boars that could contribute to a more comprehensive analysis?
  13. Discussion: could author provide a more concise discussion and better connection of data and describe how the antioxidant related data are related to the semen quality.

Comments on the Quality of English Language

According to my English knolwedge, the English is fine but could be better with some minor revision. 

Author Response

The study addresses a important aspect of sperm preservation, seeking to identify the optimal post-thaw storage conditions to extend sperm survival time without compromising quality. While the manuscript presents good findings, there are several questions and concerns that require clarification and further information.

Point 1. Osmolarity of Extender: Could the authors provide information on the osmolarity of the extender used in the study before the addition of glycerol?

Response 1: Before the addition of glycerol, the osmolarity of Tris-citric acid-glucose extender is 314 mOsmol/kg. The information has been added to the text in 2.1 reagents and media.

Point 2. Semen Collection Details: a. How many semen collections were performed per week? b. What was the distance between each semen collection? How raw semen was processed? How was it transported to the lab? What was the temperature of transportation?

Response 2: The boar involved in this study were conventionally used for AI doses production. The frequency of semen collection is two time per week, which is a regular practice in the boar stud. The interval between two collections is 3 days. The raw semen samples were collected using a hand-gloved method and diluted 1:1 in volume with BTS, whose quality was thereafter evaluated. Then the semen samples were processed for freezing. In fact, the semen freezing experiment was performed in the boar stud. The semen samples were collected and immediately utilized for freezing. The transportation of semen from collection room to AI lab was within 5 min at room temperature.

Point 3. Morphological Assessment Protocol: How was the morphology measured as the initial evaluation of semen? Please provide details on the protocol used. Was it expanded morphology?

Response 3: The initial evaluation of sperm morphology was conducted in the boar stud. Sperm viability and abnormality were evaluated using an eosin-nigrosine staining method under microscope. At least 300 sperm cells were counted. The abnormality of sperm includes proximal cytoplasmic drops, curled tail, folded tail, separate head and tail. The morphologic evaluation was used to pre-select good-quality semen samples for freezing.

Point 4. Dilution Medium for Initial Assessment: When semen was diluted 1:1 with BLT for the initial assessment, did the dilution medium contain glycerol and egg yolk, or was it a basic medium? What was the ratio of dilution and at what temperature?

Response 4: BTS was used as a basic medium to dilute semen at the beginning, which contains no glycerol neither egg yolk. After semen collection, semen samples were diluted with BTS in a ratio of 1:1 in volume at the same temperature of 32 ℃. This information has been provided in the text.

Point 5. Concentration of Semen per Straw: What was the concentration of semen per straw used in the study? What was the final volume of semen per dose?

Response 5: The final concentration was 1.0 × 109 sperm/mL and the final volume was 0.5 mL for each straw.

Point 6. Use of Pooled Semen: Why did the authors choose to use pooled semen from each replicate? Was there a specific reason for not assessing the individual impact of each boar separately? How did you choose the semen per pool?

Response 6: There are five ejaculates from five boars involved in this study. To avoid variations in boar individuals and semen ejaculates, straws from each boar were thawed and mixed with straws from another boar to make a pooled semen sample. The five boars were previously selected by the boar stud in terms of their sperm freezability. All of them showed good freezability and thus chosen to be used for this study. The variations in boar races and individuals were confirmed by previous reports. Here in this study we were focusing on the effect of post-thaw storage on sperm lifespan. Therefore, semen samples were pooled to avoid the variation mentioned. In fact, straws from the five boars were randomly taken out of the liquid nitrogen tank and thawed. Then the thawed semen was mixed one and another without repetition.

Point 7. PBS Composition: a. Was the PBS magnesium-free? b. How about calcium content? c. What was the pH of the PBS?

Response 7: The PBS used in this study was composed by NaCl, KCl, KH2PO4, EDTA, Na2HPO4, Penicilin G and Streptomycin Sulfat. The pH was 6.8-6.9, osmolarity 280-300 mOsmol/kg. There is no magnesium neither calcium.

Point 8. Sperm Motion Analysis (SCA): During sperm motion analysis in section 2.3, how many frames per second were recorded? Please provide all the settings used for the SCA

Response 8: The CASA (ISASV1®; Proiser R+D, Paterna, Spain) used in this study was configured: 25 frames recorded per second, particle area 10-80 μm2, velocity 20 μm/s<slow<25 μm/s <medium<45 μm/s <rapid, VSL>=40 μm/s and STR>45% as progressive sperm. The information has been added in the text.

Point 9. Flow Cytometry Assessment: a. What served as the positive control in the flow cytometry assessment? b. Could the authors provide technical information, including wavelengths, excitation and emission details, bandpass information, and longpass information for all individual fluorescent probes used?

Response 9: For the flow cytometric analysis, negative control (without the target staining) was set in all the parameters evaluated. Positive control was set in ROS measurement with TBH treatment. A flow cytometer (CytoFLEX S, Beckman Coulter Inc, Brea, CA, USA) was employed for analysis of sperm viability, mitochondrial membrane potential, intracellular ROS production and sperm apoptotic changes. The optimal voltage for fluorescence detection for this flow cytometer (loaded with three lasers) was configured as FSC=502V, SSC=504V. Hoechest 33342 was used to gate debris from cell populations. A minimum of 10,000 cells were calculated. PNA-FITC (excitation wavelength=495nm, emission wavelength=520nm) was used to separate sperm populations with intact and damaged acrosome membrane. Propidium iodide (excitation wavelength=488nm, emission wavelength=630nm) was used to separate sperm populations with intact and damaged plasma membrane. Mitotracker Deep red 633 (excitation wavelength=622nm, emission wavelength=648nm) was used to separate sperm populations with high and low mitochondrial membrane potential. CM-H2DCFDA (excitation wavelength=488nm, emission wavelength=525nm) was used to separate sperm populations with high and low intracellular ROS production. Annexin V-FITC (excitation wavelength=495nm, emission wavelength=520nm) was used to detect apoptotic changes in sperm cells.

Point 10. Malondialdehyde (MDA) Standard Curve: Regarding the MDA, could the authors provide information on the standard curve used in the study?

Response 10: The malondialdehyde (MDA) contents indicating lipid peroxidation level were assayed using a commercial kit (A003-1-2, Nanjing Jiancheng, China), following the manufacturer's instructions. The MDA contents were calculated based on the equation: MDA (nmol/mL)=OD value/OD value of standard MDA (10 nmol/mL) *dilution ratio. According to the instructions of this commercial kit, there is no need to make standard curve for the calculation of MDA contents.

Point 11. Lack of Fertility Data: One major concern is the absence of fertility data in the paper. Is it possible for the authors to include fertility data, such as pregnancy outcomes?

Response 11: Thank you for your valuable suggestions. It will be ideal to have field data in this study to confirm that the sperm with extended lifespan would achieve good fertility results. Actually, the authors were planning to include this part during when the experiment was conducted. However, due to the ASF breakout and the biosecurity measures taken in pig farms, it was not possible to perform the artificial insemination with frozen-thawed boar semen. Apart from this, the main purpose of this study was to seek the best post-thaw storage method to extend sperm lifespan and thus provide time flexibility for future use in artificial insemination.

Point 12. Additional Data Request: a. Is there post-thaw morphology information that could be added to enhance the paper? b. Do the authors have data for individual boars that could contribute to a more comprehensive analysis?

Response 12: The morphology data of post-thaw boar sperm has been included in the original data EXCEL FILE 4 in the supplementary files. The post-thaw sperm quality data was included for each of the five boars involved in this study. This was to give an original data to show the freezability of those ejaculates. To avoid variations in boars and ejaculates, pooled semen was made and evaluated in post-thaw section.

Point 13. Discussion: could author provide a more concise discussion and better connection of data and describe how the antioxidant related data are related to the semen quality.

Response 13: The authors really appreciate your valuable comments. We have discussed the results and explain them by connecting the quality with apoptotic levels and lipid peroxidation levels, which is the key findings of the present study. The modifications according your comments have been made in the text.

Round 2

Reviewer 1 Report

Comments and Suggestions for Authors

The authors already corrected all the figure legends.

Reviewer 2 Report

Comments and Suggestions for Authors

The manuscript has been corrected upto the mark; therefore, this may be considered for publication